# Towards an Improved Understanding of the Effects of Elevated Progesterone Levels on Human Endometrial Receptivity and Oocyte/Embryo Quality during Assisted Reproductive Technologies

**DOI:** 10.3390/cells11091405

**Published:** 2022-04-21

**Authors:** Nischelle R. Kalakota, Lea C. George, Sara S. Morelli, Nataki C. Douglas, Andy V. Babwah

**Affiliations:** 1Department of Obstetrics, Gynecology and Reproductive Health, Rutgers Biomedical and Health Sciences, Rutgers University, Newark, NJ 07107, USA; lcg84@njms.rutgers.edu (L.C.G.); morellsa@njms.rutgers.edu (S.S.M.); nd537@njms.rutgers.edu (N.C.D.); 2Center for Immunity and Inflammation, Rutgers Biomedical and Health Sciences, Rutgers University, Newark, NJ 07107, USA; 3Department of Pediatrics, Robert Wood Johnson Medical School, Rutgers Biomedical and Health Sciences, Rutgers University, Newark, NJ 07107, USA; avb58@rwjms.rutgers.edu

**Keywords:** assisted reproductive technology, infertility, implantation failure, endometrial receptivity, progesterone, estrogen

## Abstract

Ovarian stimulation is an indispensable part of IVF and is employed to produce multiple ovarian follicles. In women who undergo ovarian stimulation with gonadotropins, supraphysiological levels of estradiol, as well as a premature rise in progesterone levels, can be seen on the day of hCG administration. These alterations in hormone levels are associated with reduced embryo implantation and pregnancy rates in IVF cycles with a fresh embryo transfer. This article aims to improve the reader’s understanding of the effects of elevated progesterone levels on human endometrial receptivity and oocyte/embryo quality. Based on current clinical data, it appears that the premature rise in progesterone levels exerts minimal or no effects on oocyte/embryo quality, while advancing the histological development of the secretory endometrium and displacing the window of implantation. These clinical findings strongly suggest that reduced implantation and pregnancy rates are the result of a negatively affected endometrium rather than poor oocyte/embryo quality. Understanding the potential negative impact of elevated progesterone levels on the endometrium is crucial to improving implantation rates following a fresh embryo transfer. Clinical studies conducted over the past three decades, many of which have been reviewed here, have greatly advanced our knowledge in this important area.

## 1. Introduction

During the menstrual cycle, the estrogen-primed endometrium becomes receptive to embryo implantation under the influence of progesterone and cAMP. Progesterone levels are dynamically regulated across the menstrual cycle [1,2] (Figure 1). At the start of the cycle, during the follicular phase, serum progesterone is present in relatively modest but constant levels at <1 ng/mL. In the early follicular phase, circulating progesterone is largely of adrenal origin, whereas the ovaries are the main source of progesterone in the late follicular phase [3]. Following ovulation, which occurs 34–36 h after the luteinizing hormone (LH) surge, the dominant follicle undergoes rapid luteinization (the transformation of LH receptor-expressing granulosa cells into luteal cells) within a few hours (Figure 1). This results in the formation of the corpus luteum (CL) (Figure 1). Within luteal cells, cholesterol is converted to progesterone and, following ovulation, serum progesterone levels rise to over 15 ng/mL in the mid-luteal phase (Figure 1) [1,2,4,5].

Progesterone secretion in the luteal phase is episodic, and in the mid- and late luteal phase its levels are correlated closely with the pulsatile release of LH, peaking during the mid-luteal phase and declining rapidly in the late luteal phase [4] (Figure 1). Progesterone, signaling via its cognate receptors, PRA and PRB, has profound effects on the expression of the endometrial transcriptome, and thereby regulates the differentiation of the epithelial and stromal compartments, resulting in the development of the secretory endometrium (Figure 1). In the mid-luteal phase, mainly driven by progesterone, the secretory endometrium becomes receptive to embryo implantation [5,6,7,8,9,10,11,12,13,14,15]. The period of endometrial receptivity, referred to as the window of implantation (WOI), spans 30–36 h, occurring between day 6 (LH + 6) and 9 (LH + 9) after the LH surge in natural cycles (Figure 1) or between day 4 (P + 4) and 7 (P + 7) after progesterone administration in hormonal replacement therapy (HRT) cycles [16].

In ovarian stimulation (OS) cycles, in vitro fertilization (IVF) and fresh embryo transfer (ET) are associated with reduced embryo implantation and pregnancy rates relative to rates after frozen-thawed ET. A robust response to the gonadotropins used for OS is associated with supraphysiological levels of estradiol [17,18,19,20,21,22,23,24] and a premature rise in progesterone levels prior to, or on the day that, human chorionic gonadotropin (hCG) is administered for oocyte maturation [23,24,25]. Although there are different threshold values defining elevated progesterone, retrospective studies have demonstrated that progesterone levels greater than 1.5–2.0 ng/mL are associated with reduced pregnancy rates [21,22].

The mechanism of progesterone elevation seen at the end of the follicular phase of some cycles during OS is not fully understood. One proposed theory is that in IVF cycles in which the pituitary is not downregulated, the combination of increasing LH levels and the abundance of LH receptor-expressing granulosa cells due to multiple developing follicles, result in amplified LH signaling and increased progesterone production [26,27,28]. Interestingly, in pituitary-desensitized OS cycles, the progressive increase in progesterone levels was still observed during the follicular phase [27,28,29,30]. This observation led to the theory that the exogenous LH, present at high levels in some gonadotropin preparations (human menopausal gonadotropin (hMG)), stimulated granulosa cell-dependent production of progesterone. This theory was supported by studies showing a time-dependent relationship between hMG administration and an increase in progesterone levels [31]. However, it was found that, in IVF cycles, when OS was induced using purified follicle-stimulating hormone (FSH) preparations (containing less than 1% LH), a similar premature rise in progesterone during the late follicular phase also occurs [32,33,34]. Together, these studies show that the trigger inducing elevated progesterone levels during OS originates in the ovary. This idea was further strengthened by the observation that even adrenal suppression in an OS cycle did not prevent the premature rise in progesterone [35].

The premature rise in progesterone levels in the follicular phase during OS is associated with reduced implantation and pregnancy rates [21,22], believed to be due to an unreceptive endometrium and potentially poor oocyte/embryo quality. Indeed, studies report that, in gonadotropin-stimulated cycles, during the WOI, the endometrium is histologically advanced, and this is coupled to a dysregulation in the expression of genes that regulate embryo implantation [36,37,38]. With respect to the impact on oocyte/embryo quality, there are contrasting reports regarding the effect of elevated progesterone. Some reports suggest a negative effect [39,40] while the majority of studies reported minimal to no effects [27,41,42,43]. Taken together, these findings suggest that the poor IVF outcomes associated with high progesterone levels are the result of a negatively affected endometrium rather than poor oocyte quality. Specifically, it is proposed that OS in high responders advances the WOI, resulting in embryo–endometrial asynchrony and implantation failure [44,45,46,47].

This article reviews published clinical studies from PubMed spanning the period 1994–2021. Studies were first identified using keyword search terms that included: ovarian, stimulation, progesterone, gene expression, oocyte, embryo, and development. Only studies reporting a clearly defined elevated progesterone level as the single variable factor following ovarian stimulation were included in this review. Collectively, results from these studies consistently show that elevated progesterone levels alter endometrial receptivity by affecting endometrial histological development and altering the expression of the endometrial transcriptome. This article also reviews the impact of elevated progesterone levels on oocyte/embryo quality. Findings from these studies show that the impact on oocyte/embryo quality is minimal, and strongly support the assumption that, during OS, it is the negative effect of elevated progesterone on the endometrium that is responsible for reduced implantation and pregnancy rates.

## 2. The Impact of Progesterone Levels on Endometrial Histological Development and Endometrial Receptivity

### 2.1. Introduction

It was observed that, in in vitro fertilization/embryo transfer (IVF/ET) cycles, a subtle increase in progesterone levels during the follicular phase was associated with the advanced histological development of the endometrium and decreased pregnancy rates [25,48]. However, it was not established whether the premature rise in progesterone levels induced these abnormalities since the contributory roles of the pituitary-desensitizing and ovulation-inducing drugs, and/or the rapidly rising estrogen levels, could not be excluded. The goal of this section is to investigate the effect of elevated progesterone levels on endometrial histological development, and to determine the progesterone thresholds that are necessary for normal histological development. Collectively, these studies provide a clearer understanding of the effect of progesterone levels on the histological development of the endometrium and subsequent endometrial receptivity, in the absence of other confounding factors. Please refer to Table 1 for the inclusion and exclusion criteria of the studies reviewed in the following sections.

### 2.2. Assessing the Impact of Progesterone Levels on Endometrial Histological Development in an Ovarian Failure Model Using Endometrial Dating

A prospective study was conducted on 16 subjects with primary or secondary premature ovarian failure. All women had elevated FSH and LH (>50 IU/L) and low estradiol levels (<25 pg/mL). These women were administered estrogen (4 mg/day) daily during a 26-day artificial cycle, and daily intramuscular (IM) injections of progesterone (50 mg/day) starting on cycle day 15 to mimic the luteal phase [48]. Following pre-treatment, these women were randomized into two groups, Groups A and B, to receive progesterone injections in the artificial follicular phase. Group A (*n* = 8) received intramuscular (IM) progesterone injections (12.5 mg) in the follicular phase on cycle days 2 and 7, while Group B (*n* = 8) received IM progesterone injections (6.25 mg) on cycle days 3, 4, and 5. An age-matched control, Group C (*n* = 16), consisted of women who underwent the identical estradiol/progesterone protocol to induce an artificial cycle but without the follicular phase progesterone injections. Among all groups, serum estradiol and progesterone measurements, as well as endometrial biopsies (EMBs), were performed on cycle days 14 (late follicular phase) and 26 (late luteal phase) (Table 1, [48]). All biopsies were dated according to the criteria of Noyes et al. [7].

The results show that serum estradiol levels were comparable between the study and control groups, A, B, and C, on both days 14 and 26. Serum progesterone levels were also comparable between the groups on day 26, but were higher in the follicular phase of the study groups (1.9 ± 4.0 ng/mL) compared to the control group (0.2 ± 0.1 ng/mL). In the study groups, histological analysis of the cycle day 14 biopsies showed that 8 out of 16 subjects displayed a secretory phenotype in the late follicular phase. At cycle day 26, 9 out of 16 women exhibited endometrial developmental abnormalities in the late luteal phase. These abnormalities consisted of (1) both stromal and glandular cells showing an out-of-phase phenotype, (2) asynchronous glandular–stromal maturation, and (3) cases where only one of the two cell types exhibited a difference in maturation compared to controls. In summary, these findings derived from a model of ovarian failure in which an artificial cycle is induced, reveal that exogenous episodic surges of progesterone during the follicular phase impair endometrial development, which cannot be corrected by progesterone supplementation during the luteal phase.

### 2.3. Assessing the Impact of Progesterone Levels on Endometrial Histological Development in an IVF Donor Cycle Using Endometrial Dating

In a study involving an oocyte donor and embryo recipient protocol, the secretory endometrium in the donor was analyzed histologically [49]. Oocyte donors achieved pituitary desensitization with gonadotropin-releasing hormone (GNRH) agonist administration, using either daily leuprolide acetate (Lupron) injections or Depo Lupron. Following the onset of menses, OS was achieved with hMG, and the ovarian response was monitored via serum estradiol levels and transvaginal ultrasounds. Thirty-six hours after administration of hCG for final oocyte maturation, oocyte retrieval and EMB were performed (Table 1, [49]). All biopsies were dated according to the criteria of Noyes et al. [7]. Progesterone levels were measured on the day of hCG administration, as well as 1 and 2 days before hCG administration.

Histological examination of the biopsies (*n* = 25) revealed two distinct groups with respect to endometrial patterns: mixed pattern (days 14 to 15, *n* = 13) and secretory pattern (days 16 to 17, *n* = 12). In the mixed pattern, both proliferative and early secretory glands were observed, with the proliferative phenotype (subnuclear vacuoles) being predominant. Among the two groups, the duration of exposure to hMG, circulating serum estradiol on the day of hCG administration, oocyte number, serum estradiol levels per oocyte, and endometrial thickness on the day of hCG, did not differ. However, on the day of hCG administration, serum progesterone levels were significantly higher in the secretory group (1.7 ± 0.2 ng/mL) than in the mixed group (0.8 ± 0.1 ng/mL). Only 1 of the 12 subjects with a secretory endometrial pattern had serum progesterone levels <0.9 ng/mL on the day of hCG, while 3 of the 13 subjects with a mixed pattern had progesterone levels ≥0.9 ng/mL. Differences in progesterone levels were also observed among the two groups before the day of hCG. In the mixed pattern group, mean progesterone levels were 0.6 ± 0.1 ng/mL and 0.7 ± 0.1 ng/mL, 2 and 1 day(s) before hCG, respectively. In the secretory group, mean progesterone levels were 1.0 ± 0.2 ng/mL and 1.3 ± 0.2 ng/mL, 2 and 1 day(s) before hCG, respectively. Overall, as early as 2 days before hCG, 9 of the 12 subjects with a secretory endometrium already had a progesterone level ≥0.9 ng/mL.

This study clearly shows that on the day of oocyte retrieval, almost half (48%) of all OS cycles following pituitary desensitization were associated with advanced histological development of the endometrium. Interestingly, regardless of whether oocytes were derived from donors with mixed or secretory endometrial patterns, the outcomes of clinical pregnancy and delivery rates for the recipients were similar. This study demonstrates that a small increase in progesterone levels in the follicular phase induces secretory endometrial transformation, and advances histological development by at least 2 days [36]. This study was among the first to consider the impact of elevated progesterone levels on shifting the WOI, resulting in embryo–endometrial asynchrony at the time of ET. It also inspired the use of personalized medicine to aid physicians in choosing the most effective date to perform ET in artificial reproductive technology (ART) cycles [44].

### 2.4. Assessing the Impact of Progesterone Levels on Endometrial Histological Development in an IVF/ET Cycle Using Noninvasive High-Resolution Transvaginal Ultrasonography

The effect of premature elevation of progesterone on endometrial development in an OS cycle would again be documented, not by EMB and histological dating, but via noninvasive high-resolution transvaginal ultrasonography. Ultrasonography is used for assessing endometrial echogenicity, which is calculated as the extent of the submyometrial hyperechogenic transformation of the endometrium over the whole endometrial surface [50]. Ultrasonography detects changes in the development of the normal secretory endometrium [5], but operator-dependent variability often leads to poor precision and inconsistency of ultrasound measurements. To overcome this weakness, in this study, ultrasonography was coupled to a computer-assisted module for the objective analysis of ultrasound images used for monitoring endometrial histologic changes in the luteal phase following OS [50].

A prospective study was conducted on 59 IVF/ET subjects who achieved pituitary desensitization using a GNRH agonist (leuprolide acetate). This was followed by hMG administration for OS, oocyte retrieval at 36 h post hCG injection, and ET 2 days after oocyte retrieval. The luteal phase was supported with 300 mg of micronized progesterone starting on the evening of the day of ET. On the days of hCG administration, oocyte retrieval, and ET, study participants underwent transvaginal ultrasounds (Table 1, [50]). Sagittal views of the uterus were captured, then images were digitized and analyzed with a computer-assisted module developed for quantifying endometrial echogenicity and thickness. Following digitation of the uterine images, transverse cuts were performed across a representative section of the endometrial surface. This was followed by gray-level analysis on all cuts, and the average values of this analysis were graphed. Endometrial echogenicity was calculated as the ratio of the extent of the endometrial submyometrial hyperechogenic transformation relative to the entire endometrial surface. Qualitatively, if echogenicity values were greater than those of the surrounding myometrium by ≥10%, the ultrasonographic endometrial texture was considered hyperechogenic. Endometrial borders were set arbitrarily as the outer limits of the hyperechogenic myometrium–endometrium interface, and endometrial thickness was determined by measuring the greatest distance between the outer limits of the proximal and distal endometrial junctions.

Based on plasma progesterone levels on the day of hCG administration, OS cycles were divided into a low progesterone (≤0.9 ng/mL) or high progesterone (>0.9 ng/mL) group. Among the two groups, significant differences were not observed in the number of oocytes retrieved, number of embryos obtained, or the median number of embryos transferred. On the day of hCG administration, a positive correlation between estradiol and progesterone levels was observed, where plasma estradiol levels were significantly lower in the low progesterone group (2279 ± 149 pg/mL) than in the high progesterone group (2880 ± 236 pg/mL). From the day of hCG administration onwards, plasma estradiol levels declined, and estradiol levels were comparable on the day of oocyte retrieval in the low (1135 ± 121 pg/mL) and high (1213 ± 144 pg/mL) progesterone groups, as well as on the day of ET in the low (1008 ± 92 pg/mL) and high (1141 ± 90 pg/mL) progesterone groups.

Based on the results of the computer-assisted ultrasound evaluation, in the low vs. high progesterone groups, endometrial thickness was similar on the days of hCG administration (10.2 ± 0.4 mm vs. 10.2 ± 0.4 mm), oocyte retrieval (9.7 ± 0.6 mm vs. 10.7 ± 0.4 mm), and ET (9.4 ± 0.4 mm vs. 9.9 ± 0.3 mm). Thus, an effect of plasma estradiol levels on endometrial thickness was not observed on the days of hCG administration, oocyte retrieval, or ET. On the day of hCG administration, endometrial echogenicity values were similar in the low (0.40 ± 0.16) and high progesterone groups (0.41 ± 0.19). After hCG administration, the echogenicity values increased progressively in both groups on the days of oocyte retrieval and ET, but the increase was greater in the high progesterone group (day of oocyte retrieval, 0.70 ± 0.16; day of ET, 0.90 ± 0.23) than in the low progesterone group, (day of oocyte retrieval, 0.63 ± 0.17; day of ET, 0.78 ± 0.21). Additionally, a positive correlation between plasma progesterone levels and endometrial echogenicity values was observed on the days of oocyte retrieval and ET. No relation between plasma estradiol levels and endometrial echogenicity was noted.

The results from this study show that the increase in the endometrial echogenicity following hCG administration in OS cycles is accelerated in subjects who display premature progesterone elevation. Based on a previous study demonstrating that endometrial echogenicity reflects endometrial histology [5], these results support the observation that, in IVF/ET cycles, a premature elevation in progesterone level is associated with a faster secretory transformation of the endometrium, and hence the advanced development of the receptive endometrium. This study also highlights the potential for using ultrasound as a noninvasive alternative to EMBs for monitoring postovulatory changes of the endometrium and correlating progesterone levels with echogenicity.

### 2.5. Assessing the Impact of Progesterone Levels on Endometrial Histological Development in an IVF/ET Cycle Using Endometrial Dating

Thus far, we have seen that regardless of the condition under which endometrial histological development is assessed (using either histological dating or ultrasonography), elevated progesterone levels are consistently associated with the advanced development of the endometrium. This finding is further supported by the following study involving women undergoing OS and IVF who did not proceed to a fresh embryo transfer, and where endometrial development was assessed by endometrial dating [51].

In this prospective study, 106 women achieved pituitary desensitization with GNRH agonist (Triptorelin) administration. This was followed by OS with recombinant FSH and an hCG trigger for final oocyte maturation. After oocyte retrievals, IVF and embryo cryopreservation were performed. None of the women underwent ET during this cycle. Progesterone luteal support was administered from the night of oocyte retrieval until the day of EMB. During the study period, estradiol and progesterone were measured on the day of hCG administration (12–14 h before injection) and 12–14 h after hCG administration (hCG + 1). An EMB was performed 7 days after hCG administration (Table 1, [51]). All biopsies were dated according to the criteria of Noyes et al. [7]. Glandular–stroma asynchrony was defined as ≥4-day difference between the development of glandular and stromal cells. In this study, the authors defined high progesterone levels as ≥1.7 ng/mL on the day of hCG administration, and ≥9.5 ng/mL on the day after hCG administration (hCG + 1).

The results show that 58 subjects had high progesterone levels on the day of hCG administration and on hCG + 1. However, only progesterone levels on hCG + 1 showed an association with histological staging. Specifically, the results show that endometrial development in women with a high progesterone level (*n* = 58) was more advanced than that of women with normal progesterone (*n* = 48). This advanced development was in the range of 0.3 to 1.0 days, based on assessment conducted by two independent investigators. In addition, glandular–stroma asynchrony was more frequent in women with normal progesterone compared to women with high progesterone. Logistic regression analysis revealed that there was no association between estradiol levels on the day of hCG, or on hCG + 1, and the histological staging of biopsies collected 7 days after hCG administration.

Overall, in this study [51], the investigators found that high progesterone in the early secretory phase (hCG + 1) was associated with advanced histological endometrial development (both glandular and stromal compartments) on hCG + 7. Alternatively, normal progesterone levels (<1.7 ng/mL on the day of hCG administration and <9.5 ng/mL on hCG + 1) were associated with glandular–stroma asynchrony. These unexpected findings might, in part, be due to the time at which the biopsies were performed relative to the progesterone measurements. Progesterone was measured on the day of hCG administration and on hCG + 1, while biopsies were performed on hCG + 7. Despite the use of regression analysis to determine the effect of early progesterone levels on luteal phase endometrial development, the relatively long intervening period between progesterone measurements and the day of biopsy could have prevented an accurate interpretation of the effect of elevated progesterone on endometrial development. In two of the studies reviewed earlier in this section [48,50], progesterone measurements and biopsies were conducted at the same time point, while in another study [49], progesterone measurements were made approximately 2 and 3 days into the follicular phase before the biopsies were conducted in the early luteal phase. Taken together, these investigators found that elevated progesterone levels in the follicular phase were associated with the histological advancement of the endometrium and a higher incidence of glandular–stroma asynchrony.

### 2.6. Assessing the Impact of Progesterone Levels on Endometrial Histological Development in Healthy Women Undergoing Experimentally Modeled Endometrial Cycles Using Endometrial Dating

To further establish which concentrations of secretory phase progesterone are associated with altered endometrial structure and/or function, the following study [52] employed experimentally “modeled endometrial cycles” that provided greater control over estradiol and progesterone levels during the period of analysis [52,53]. In this study, investigators examined the effects of four different doses of exogenously administered progesterone concentrations. Three of these doses resulted in endogenous progesterone levels that are typically observed during the luteal phase, while the fourth resulted in a level that was lower than 3 ng/mL, i.e., the concentration associated with the postovulatory luteal phase in natural menstrual cycles.

In this prospective study, ovulatory women achieved pituitary desensitization using a GNRH agonist (leuprolide acetate). After confirming effective downregulation (serum estradiol < 40 pg/mL and ovarian follicles <10 mm), subjects received transdermal estradiol (0.2 mg/day) for 20 consecutive days. After 10 days of estradiol treatment, subjects underwent a transvaginal ultrasound to ensure endometrial thickness was at least 7 mm. Subjects were then randomized to receive one of four progesterone doses (2.5, 5.0, 10.0, or 40.0 mg) administered as a daily intramuscular (IM) injection. Except for the administration of varying doses of progesterone, the induced artificial cycles were identical to treatment protocols utilized to prepare the endometrium for donor embryo transfers. Healthy women undergoing natural cycles served as the control group. Serum progesterone concentrations in the modeled cycles were measured 2 to 3 h after injection (peak) and 1 to 2 h before injection (trough) on two separate occasions between 3 and 10 days of progesterone treatment. In the natural cycle control group, EMBs (*n* = 10) were performed 10 days after the mid-cycle urinary LH surge. In the corresponding modeled cycles, biopsies were performed on progesterone day 10 on subjects receiving 2.5 mg (*n* = 6), 5 mg (*n* = 6), 10 mg (*n* = 12), or 40 mg (*n* = 12) of progesterone daily (Table 1, [52]). All biopsies were dated according to the criteria of Noyes et al. [7].

The results show that the mean peak and trough serum progesterone concentrations were different among the groups. In subjects receiving the highest dose of progesterone (40 mg), the peak (18.1 ± 5.1 ng/mL) and the trough (9.4 ± 4.8 ng/mL) serum progesterone concentrations corresponded to values observed during the normal mid-luteal phase. In women receiving 10 mg progesterone daily, the peak and trough serum progesterone levels were 7 ± 2.9 ng/mL and 3.3 ± 1 ng/mL, respectively. In subjects receiving 5 mg progesterone, the peak and trough progesterone concentrations were 4.2 and 2.4 ng/mL, respectively, and in subjects receiving 2.5 mg progesterone daily, peak and trough progesterone levels were 2.5 and 0.3 ng/mL, respectively. Among all samples from the natural and modeled cycles, a secretory histology was observed at all progesterone doses; however, with decreasing doses of progesterone there was an increasing frequency of delayed endometrial histologic development between the expected and observed histologic dates. In subjects receiving 5 mg of progesterone, the developmental dating varied from normal in some samples to overtly delayed in others. This suggests that the threshold of progesterone required for normal histologic development is around the lower limit of the levels normally observed across the luteal phase of natural cycles (2–4 ng/mL). In subjects receiving 2.5 mg of progesterone, which corresponded to serum progesterone levels < 3 ng/mL, histologic dating was consistently delayed by more than 3 days. The progesterone levels required for normal histologic development in these modeled cycles are like those reported in natural cycles. Histologic dating was normal in 76% and 90% of women with progesterone levels > 2 ng/mL and 4.7 ng/mL, respectively [6].

### 2.7. Conclusions

Overall, based on the studies reviewed, there is clear evidence that, during OS, a premature rise in progesterone levels advances the development of the secretory endometrium. This may lead to a higher chance of embryo–endometrial asynchrony and subsequent implantation failure [44,54]. The negative impact of embryo–endometrial asynchrony is well-documented in several studies [47,55,56,57] and has been addressed clinically with the development of the endometrial receptivity assay (ERA), which identifies the personal WOI for each woman undergoing ET [58]. This molecular tool has led to findings that >25% of patients with recurring implantation failure (RIF) of endometrial origin have a displaced or asynchronous WOI [59,60]. The endometrial receptivity analysis has proven more accurate and consistent than histological dating [7] in identifying the personalized WOI in women with RIF, and supports the hypothesis that implantation failure of endometrial origin is not an endometrial dysfunction or pathology, instead, it is the inability to synchronize the developing embryo with a patient’s individual WOI [18,61,62].

## 3. The Impact of Progesterone Levels on The Endometrial Transcriptome and Endometrial Receptivity

### 3.1. Introduction

Endometrial receptivity depends on the duration of progesterone exposure after the endometrium has been exposed to estradiol. Numerous studies have investigated the negative influence that elevated hormone levels, attained during OS, may have on endometrial receptivity. While the incidence of elevated progesterone during OS varies according to the stimulation protocol, among women using GNRH agonists, the prevalence can be as high as 35% [61]. During a natural cycle, the rise in progesterone is linked to the LH surge, which synchronizes the embryo with the total time of endometrial progesterone exposure [62]. Frozen embryo transfers have higher pregnancy rates compared to fresh transfers, likely due to estradiol/progesterone-primed cycles that are closer to the physiological conditions seen in a natural cycle [63]. While the full scope of OS-induced endometrial changes is unknown, one difference between an OS and natural cycle is in the WOI, where some studies report a delayed expression of endometrial genes that are important regulators of embryo implantation [22,39,58,64,65]. Please refer to Table 2 for the inclusion and exclusion criteria of the studies discussed in the following sections.

### 3.2. Assessing the Impact of Randomized Serum Progesterone Concentrations on Endometrial Gene Expression

The timing and concentration of progesterone exposure is critical for normal implantation. While premature elevations in this hormone can alter endometrial receptivity and histological development, a minimum concentration is also needed for successful embryo implantation. To improve implantation rates in IVF/ET cycles following OS, we need to understand and accurately quantify the amount and timing of progesterone exposure necessary for optimal endometrial development. In a previous section (“Assessing the impact of progesterone levels on endometrial histological development in healthy women undergoing experimentally modeled endometrial cycles using endometrial dating”) the study by Young et al. [52] was reviewed in the context of the effect of serum progesterone levels on endometrial histological development. That study will now be reviewed regarding the effect of progesterone concentration on endometrial gene expression. For a detailed description of the prospective study design involving 36 women allocated into four progesterone study groups, the reader is referred to the previous section. Briefly, EMBs were performed 10 days after the mid-cycle urinary LH surge (mid-secretory control group, *n* = 4) and on progesterone day 10 in subjects receiving 2.5 mg (*n* = 4), 5 mg (*n* = 3), 10 mg (*n* = 5), or 40 mg (*n* = 5) of progesterone daily (Table 2). RNA was then extracted from the EMB samples, and a microarray analysis of gene expression was performed. An analysis of variance (ANOVA) was used to identify genes with differential expression across the treatment groups.

To perform an in silico analysis of the gene expression results, the list of differentially expressed genes identified in this investigation (*n* = 497) was compared to genes identified as critical regulators of human endometrial receptivity in two previous studies [58,68]. The study by Altmäe et al. [68] identified differentially expressed genes during the mid-secretory phase on LH + 7 between infertile (*n* = 4) and fertile (*n* = 5) women, while the study by Díaz-Gimeno et al. [58] identified genes differentially expressed at the receptive phase (LH + 7) in healthy women. After identifying common genes, their expression across the menstrual cycle was examined using GEO Dataset Record GDS2052 (endometrium throughout the menstrual cycle) [69]. The expression of these genes was then grouped according to progesterone levels.

As summarized in the previous section, the results demonstrate that mean peak and trough serum progesterone concentrations were different among the groups. In subjects receiving the highest daily dose (40 mg progesterone daily), the peak (18.1 ± 5.1 ng/mL) and the trough (9.4 ± 4.8 ng/mL) serum progesterone concentrations corresponded to values observed during the normal mid-luteal phase. In women receiving the lowest daily dose of progesterone (2.5 mg), the peak (2.5 ng/mL) and the trough (0.3 ng/mL) progesterone levels were lower than the minimum concentration (3 ng/mL) observed in the luteal phase.

A total of 2275 genes were identified, and hierarchical clustering demonstrated a pattern of segregation according to progesterone dose. The controls and the 40 mg progesterone group clustered together, while the groups receiving 2.5 mg and 5 mg clustered together. The group that received 10 mg was split into multiple branches. Treatment with 2.5 mg progesterone yielded 20 downregulated genes and 147 upregulated genes compared to the treatment group receiving 40 mg of progesterone daily. Treatment with 40 mg of progesterone daily led to 26 upregulated genes and only 1 downregulated gene (vasoactive intestinal peptide receptor 2 (*VIPR2*)) when compared to the 10 mg progesterone treatment group. Differential gene expression was not specifically analyzed for the 5 mg progesterone treatment group as it was anticipated that the genes in this group would likely be included in the 2.5 mg and 10 mg treatment groups. Ingenuity pathway analysis identified two genes (*CD94* and *NKG2A*) that were upregulated in natural cycles as well as in the subjects treated with doses of 10 mg and 40 mg of progesterone, when compared to subjects treated with 2.5 mg and 5 mg of progesterone. These two genes are critical mediators of natural killer cell signaling. The findings from this investigation support the hypothesis that differences in progesterone concentrations lead to altered endometrial gene expression.

The in silico analysis identified 18 genes that overlapped with previously identified genes [58,68]. Three progesterone-regulated genes (homeobox A10, *HOXA10*; mitogen-inducible gene 6 protein, *MIG6*; and decay-accelerating factor, *DAF*) were selected and analyzed by RT-PCR. *HOXA10* was maximally expressed in subjects receiving 40 mg of progesterone daily, while neither *MIG6* nor *DAF* demonstrated differential expression, but were induced by all progesterone doses compared to proliferative phase controls. While some differentially expressed genes had monophasic changes in response to progesterone concentrations, others had multiphasic responses, such as *HOXA10*, which is an endometrial transcription factor required for embryo implantation. *HOXA10* expression is known to be reduced in disorders that negatively impact implantation [70], thereby suggesting that progesterone concentrations affect endometrial receptivity.

### 3.3. Elevated Progesterone Levels on the Day of hCG Trigger Alters Endometrial Gene Expression during Ovarian Stimulation

Similar to the findings above, an investigation performed by Labarta et al. [66] also revealed that elevated serum progesterone levels, on the day of hCG trigger, can significantly alter the gene expression profile of the endometrium. As discussed earlier, during OS, an elevation in progesterone levels can be seen at the end of the follicular phase. This premature rise is not seen in natural cycles, and is presumed to have a negative impact on embryo implantation [62].

The study performed by Labarta et al. [66] was a single-center prospective cohort study conducted between April 2007 and July 2009. Twelve women, ages 18–35 years, were included in the study; they were separated either into a group that consisted of subjects with a progesterone level < 1.5 ng/mL (low progesterone) on the day of recombinant chorionic gonadotrophin (rCG) trigger (*n* = 6), or a group whose participants had progesterone levels > 1.5 ng/mL (high progesterone) on the day of trigger (*n* = 6). Each group had subjects who underwent a GNRH agonist long protocol (*n* = 3) or a GNRH antagonist multidose protocol (*n* = 3), for pituitary downregulation. Estradiol measurements on the day of rCG trigger showed that levels in both groups were not statistically different. Oocyte retrieval was performed 36 h after rCG administration. The luteal phase was supplemented with 400 mg/day of micronized progesterone, starting the day after oocyte retrieval, to simulate a cycle preparing for ET after OS. Serum progesterone levels were measured on the day of rCG trigger, and a total of 12 EMBs were collected at rCG + 7, which was considered as the WOI (Table 2). RNA was then isolated, and all samples underwent microarray analysis of gene expression. Gene expression profiles were compared using a significance analysis of microarray data (SAM). Genes with an absolute fold-change of 2.0 or greater were considered as differentially expressed. The database for annotation, visualization, and integrated discovery was used to detect activation or inactivation in biological functions or pathways [71].

A total of 140 genes were found to be differentially expressed between the low vs. high progesterone groups, 64 were upregulated, while 76 were downregulated. However, when using the Rank product, a test for detecting differentially expressed genes, more genes were found to be dysregulated, with 209 downregulated and 262 upregulated in the high progesterone group. When comparing the list of dysregulated genes to the 25 WOI genes known to be related to endometrial receptivity [72], 13 showed dysregulation in women with high progesterone levels (7 upregulated and 6 downregulated genes). All 13 genes showed higher fold changes than those observed in the natural cycle, and 8 of these 13 genes have putative progesterone response elements (PRE) in their regulatory sequences. Through PCA analysis, the 140 differentially expressed genes were found to cluster into two distinct groups corresponding to low progesterone (<1.5 ng/mL) and high progesterone (>1.5 ng/mL). Analyses of the biological processes, molecular functions, and KEGG pathways associated with the differentially expressed genes revealed differential expression among genes involved in cell adhesion (e.g., *LAMA4* and *ITGB2)*, as well as developmental (e.g., *SMAD9* and *RND3*) and immune processes (e.g., *ILIB* and *TLR5*).

This study by Labarta et al. [66] demonstrates that gene expression profiles in endometrial samples collected from women with prematurely elevated progesterone differ from those collected from women without premature progesterone elevation. These findings once again support the hypothesis that endometrial receptivity is altered by early exposure to elevated progesterone levels, and may be the cause of lower implantation rates following OS.

### 3.4. Microarray Analysis of Endometrial Gene Expression in OS Cycles with a Premature Elevation in Progesterone Levels Uncovers the Dysregulated Expression of Cell Cycle Genes in the Pre-Receptive Phase

Recent studies have demonstrated that progesterone levels on the day of hCG administration in OS cycles can lead to epigenetic modification of the endometrium during the peri-implantation period [73]. This alteration in gene expression is thought to disrupt endometrial receptivity and lead to decreased pregnancy rates following OS. In a study conducted on women undergoing OS, Haouzi et al. [67] investigated the impact of a premature elevation in serum progesterone levels (>1.5 ng/mL), on the day of hCG trigger, on gene expression profiles during the pre-receptive (hCG + 2) and receptive (hCG + 5) secretory stages. Data from this experimental cohort were compared to the gene expression profiles obtained from women with normal progesterone levels (<1.5 ng/mL) on the day of hCG trigger.

The study population included 15 women, ages 31 ± 3 years, who were referred for intracytoplasmic sperm injection (ICSI) due to male infertility. All subjects had normal serum gonadotropins and estradiol levels on day 3 of OS, utilizing either a GNRH agonist or antagonist protocol, as well as on the day of hCG trigger. EMBs were obtained on the day of oocyte retrieval (hCG + 2) and at the time of embryo transfer (hCG + 5) (Table 2). Following EMB collection, RNA was extracted to perform microarray analysis of gene expression using the Human Genome U133 Plus 2.0 Array (Affymetrix), which provides complete coverage of the human genome for the analysis of over 47,000 transcripts. Subjects were divided into groups according to serum progesterone concentration on the day of trigger (normal progesterone group, *n* = 7; high progesterone group, *n* = 8). The number of subjects undergoing a GNRH agonist protocol was similar in both groups (*n* = 2 per group). Gene expression profiles were then compared at both time points (hCG + 2 and hCG + 5) to identify which genes were differentially expressed between endometrial samples from subjects with normal and high progesterone levels.

A total of 6084 and 6130 genes were expressed at both secretory stages in the normal and high progesterone groups, respectively. SAM identified 1477 and 233 genes that were differentially expressed between hCG + 2 and hCG + 5 in the normal and high progesterone groups, respectively. A total of 212 genes were found to be exclusively modulated in the high progesterone group between the pre-receptive (hCG + 2) and receptive (hCG + 5) stage, 50 of which are involved in the cell cycle. Several of these genes are members of the cell division cycle family (*CDC20*, *CDC25C*, *CDCA1*, *CDCA2*, *CDCA5*, *CDCA8*), cyclins (*CCNB1*, *CCNB2*), and kinesins (*KIF4A*, *KIF11*, *KIF15*, *KIF23*). To assess the biomarkers of endometrial receptivity, an endometrial receptivity predictor list of 54 genes was used for unsupervised clustering of the endometrial gene expression profiles at the pre-receptive and receptive stages. The most highly expressed predictor genes (*n* = 13) were validated using RT-qPCR, and a significant difference was seen in the expression of *CD68* and *KRT80* between the two progesterone groups.

The findings from this investigation suggest a transcriptomic shift with high serum progesterone levels, resulting in fewer (*n* = 233) genes being differentially expressed between hCG + 2 and hCG + 5 vs. the number of differentially expressed genes found in the normal serum progesterone group (*n* = 1477). The difference in gene expression suggests that endometrial maturation is accelerated during the early secretory phase among subjects in the high progesterone group. Many of the downregulated genes in the high progesterone group were found to be involved in cell cycle functions; thus, the proposed acceleration in endometrial maturation may be a result of high serum progesterone levels altering cell growth and proliferation in the endometrium. However, the alteration of the endometrial transcriptome seen in patients with high progesterone did not seem to affect endometrial receptivity during the window of implantation. Quantitative RT-PCR analysis of known endometrial receptivity biomarkers (known to be upregulated during the WOI) demonstrated similar or higher expression levels in patients with high progesterone when compared to normal controls. The gene expression changes seen in this investigation only point to abnormally accelerated endometrial maturation during the pre-receptive secretory phase without significant alterations in endometrial receptivity at the window of implantation.

### 3.5. Examining the Effects of Elevated Progesterone Levels on DNA Methylation and Endometrial Gene and Protein Expression during Ovarian Stimulation

The incidence of elevated progesterone during OS varies according to the stimulation protocol, and among women using GNRH agonists, the prevalence can be as high as 35% [61]. As demonstrated by the studies described above, endometrial histology and endometrial gene expression are both altered by elevated progesterone levels in OS cycles. Elevated progesterone levels on the day of hCG administration have also been associated with increased endometrial DNA methylation [73]. To investigate the effect of elevated progesterone on epigenetic modifications and gene expression, Xiong et al. [61] studied the effects of high progesterone levels on DNA methylation and the gene expression profiles of endometrial adhesion molecules during the WOI. This study focused on genes encoding the adhesion proteins, mucin 1 *(MUC1*), cadherin 1 (*CDH1*), and β catenin (*CTNNB1*), because of their importance in human endometrial receptivity and embryo implantation [13,61,74,75,76,77].

A total of 40 subjects were recruited for the study, and were divided into two groups: high progesterone level (≥1.7 ng/mL, *n* = 20) and normal progesterone level (<1.7 ng/mL, *n* = 20) on the day of hCG trigger during OS cycles. This cut-off was based on previous literature, which determined 1.7 ng/mL on the day of trigger to be the 90th percentile of serum progesterone levels derived from over 1400 fresh ET cycles [78]. The subjects chosen for this investigation were ovulatory women of ages 25–40 years, who were undergoing OS with IVF for tubal or male factor infertility. The subjects underwent OS with a GNRH agonist, followed by hCG administration to induce final oocyte maturation. Serum progesterone levels were measured every 2–4 days in the follicular phase during OS, and on the day of hCG trigger (Table 2). All 40 women underwent a freeze-all approach rather than fresh ET, due to multiple factors, including risk of ovarian hyperstimulation syndrome (OHSS), desire for preimplantation genetic testing, or personal reasons. Endometrial biopsies were collected on hCG + 7, and the tissue was subsequently used to study gene (by qPCR) and protein (by immunohistochemistry (IHC)) expression of endometrial adhesion molecules and DNA methyltransferases (DNMTs). mRNA expression and semi-quantitative IHC analyses were compared using a Student’s *t*-test.

This study shows that DNMTI1 and DNMT3B were mainly expressed in the nuclei of luminal and glandular cells, and that DNMTI1 was also expressed in the nuclei of some stromal cells. While DNMTI1 and DNMT3B expression was seen in the epithelium of both the normal and high progesterone groups, DNMT3B expression was significantly higher in the high progesterone group. To quantify the DNA methylation status of the endometrial adhesion molecules, the methylation of CpG sites on the promoter regions of *MUCI, CDH1*, and *CTNNB1* was compared between the high progesterone and control groups. No significant difference was seen in the DNA methylation of *MUCI* between groups; however, for *CDH1* and *CTNNB1*, the overall methylation at CpG sites in the high progesterone group compared to the normal progesterone control group was 11 vs. 9 sites and 12 vs. 11 sites, respectively. The study, however, does not report whether these values in the high vs. normal progesterone groups were significantly different. To determine whether gene expression was altered by DNA methylation, *MUCI, CDH1* and *CTNNB1* mRNA and protein expression was analyzed. Interestingly, it was found that only *CDH1* and *CTNNB1* mRNA levels correlated negatively with DNA methylation levels; correlation with *MUC1* mRNA levels was not observed. All proteins were expressed in the luminal and glandular epithelium; however, protein levels were significantly lower in the high progesterone group compared to the normal control group. mRNA expression levels were also similar, with significantly lower expression observed in the high progesterone group.

This study by Xiong et al. [61] was the first to investigate the relationship between elevated progesterone levels on the day of hCG administration and altered DNA methylation and gene expression of adhesion molecules in the WOI. In a previous investigation by Xiong et al. [73], it was found that a high progesterone level on the day of hCG administration (during OS) is associated with epigenetic modification of the endometrium via an upregulation of 5-methylcytosine (5-mc), a marker for DNA methylation. However, in that study, DNA methylation status and its effect on endometrial receptivity under high progesterone levels was not determined. In this more recent study, Xiong et al. [61] measured the expression of DNMT1 and DNMT3B, enzymes responsible for the maintenance and deposition of DNA methylation [73]. This study demonstrated that the expression of DNMT3B, which is responsible for de novo methylation, was increased in the endometrium of women with elevated progesterone levels (≥1.7 ng/mL). However, while only *CDH1* and *CTNNB1* promoter sequences were hypermethylated in the high progesterone group, the mRNA and protein levels of *CDH1*, *CTNNB1*, and *MUC1* were decreased. Since protein expression of DNMT1, which maintains methylation at hemimethylated CpG sites, was similar between the high progesterone and normal control groups, the findings suggest that high progesterone levels may only alter de novo methylation via DNMT3B. In summary, the findings from this study reveal that high progesterone levels during the WOI are associated with DNA hypermethylation and a low expression of endometrial *CDH1* and *CTNNB1*. The reduced expression of these adhesion molecules may offer a novel and plausible mechanism underlying the reduced implantation rates observed in fresh ET following OS.

Successful implantation requires the synchronous development of a receptive endometrium and an implantation-competent embryo. Endometrial adhesion molecules play a key role in this process. The aberrant methylation, and the subsequent reduction in gene expression, observed in endometria exposed to elevated progesterone levels might underlie the impaired endometrial receptivity seen in OS cycles. Interventions to correct this epigenetic change may offer the ability to improve implantation rates in IVF/ET cycles following OS.

### 3.6. Limitations

Although the studies discussed above demonstrate an association between progesterone levels and endometrial transcriptomic changes, there are limitations that need to be addressed. The first limitation is the varying thresholds used when defining an elevated progesterone level. Two of the studies described above utilized a level of >1.5 ng/mL [66,67]. In contrast, a third study [61] used a threshold of 1.7 ng/mL, based on previous literature that determined this value on the day of hCG trigger to be the 90th percentile of serum progesterone levels during fresh ET cycles [78]. Additionally, these studies rely on one to two serum hormone measurements, which would not account for the possibility of pulsatile progesterone secretion [52]. Another noteworthy limitation is the discrepancies between the types of OS protocols utilized (GnRH agonist vs. antagonist) as well as the variation in patient demographics, including body mass index, medical conditions, menstrual regularity, and infertility history. This variation is largely due to discrepancies in exclusion/inclusion criteria across studies. Despite the limitations described, the findings from these studies clearly demonstrate an important association between elevated progesterone levels and endometrial gene expression, which offers a plausible explanation for decreased endometrial receptivity following OS.

## 4. The Impact of Elevated Progesterone Levels on Oocyte and Embryo Quality

### 4.1. Introduction

Based on the current understanding of the pathophysiology of the menstrual cycle and early pregnancy, premature progesterone elevation could impact not only the receptivity of the endometrium, but also the quality of the oocyte retrieved, and embryos created, after OS. Studies examining this relationship have conflicting findings, particularly related to oocyte and embryo quality. This section of the review will highlight sentinel studies that show mixed results. Please refer to Table 3 for the inclusion and exclusion criteria of the studies discussed.

### 4.2. Oocyte and Embryo Quality

To understand the net effects of premature progesterone elevation on the oocyte, and dissociate the effects of premature progesterone elevation on the endometrium, many investigators assessed cycles with donor oocytes or frozen embryo transfers. Findings from these investigations propose that cycles with premature progesterone elevation have similar, if not increased, numbers of oocytes retrieved, and more mature oocytes [79,80,81,82,83,84]. Interestingly, studies that investigated the effect of premature progesterone elevation on fresh ET cycles demonstrated similar results [29,44,85].

For example, the lack of effect of elevated progesterone levels on oocyte quality was clearly revealed in a retrospective study that analyzed clinical data from 68 oocyte donors and 68 oocyte recipients with ovarian failure [79]. The donor cycle began with pituitary desensitization using a GNRH agonist (leuprolide acetate), followed by hMG OS and an hCG trigger for final oocyte maturation. Follicle aspiration was followed by IVF/ET 48 h after aspiration. Oocyte recipients also underwent pituitary desensitization using a GNRH agonist (leuprolide acetate) followed by an estrogen/progesterone hormone replacement protocol. Among the donors, 21 women exhibited elevated serum progesterone levels ≥1.1 ng/mL on or before the day of hCG administration; a normal progesterone level was defined as a serum progesterone level of ≤0.9 ng/mL throughout the follicular phase. A comparison of donors from the elevated and normal progesterone groups revealed there to be no difference in estradiol levels during the periovulatory period and on day 3 after hCG administration (Table 3). When compared to normal cycles, the cycles with premature progesterone elevation had higher serum progesterone levels on the day before hCG (1.1 ± 0.7 vs. 0.6 ± 0.3), the day of hCG (1.3 ± 0.2 vs. 0.6 ± 0.2), and the day after hCG (6.0 ± 2.7 vs. 3.7 ± 1.5). However, progesterone levels were similar at 3 days after hCG (0.3 ± 0.2 vs. 0.4 ± 0.2 ng/mL) for both cycle conditions. Oocyte recipients, like the donors, also had similar estradiol levels before progesterone administration. Additionally, based on similar fertilization rates, polyspermia rates, and relative embryo quality (not specifically defined), the results show that oocyte quality was similar in women receiving oocytes from donors with and without elevated progesterone. Therefore, the investigators concluded that elevated progesterone had no effect on oocyte quality, and any negative impact of elevated progesterone on pregnancy rates in OS cycles is due to an adverse effect on the endometrium.

The impact of premature progesterone elevation on the fertilization potential of retrieved oocytes was analyzed in OS cycles resulting in IVF with frozen and fresh ET cycles. Again, those with premature progesterone elevation exhibited similar or increased fertilization rates and a similar number of euploid embryos, embryos transferred, and embryos cryopreserved [27,79,80,82,83,84,85,86,87]. To better understand the impact of premature progesterone elevation on the quality of these embryos, various embryo development rates were analyzed. No differences were observed in the duration of embryo culture, cleavage rate, or blastocyst rate, when cycles with and without premature progesterone elevation were compared [27,42,43,82,83,85,87,88]

When investigating embryo ploidy rates, Neves et al. [87] conducted a multicenter retrospective study of 1495 IVF cycles with ICSI that underwent preimplantation genetic testing for aneuploidy (PGT-A). A GNRH antagonist protocol (recombinant FSH, hMG) for OS was used, followed by an hCG trigger for oocyte maturation. Embryos were cultured and underwent a freeze-all strategy after testing (Table 3). Subjects were divided into two groups based on serum progesterone level on the day of hCG trigger: ≤1.50 ng/mL (*n* = 1328) and >1.50 ng/mL (*n* = 167). Interestingly, in the progesterone elevation group, the number of euploid embryos was significantly higher (*p* = 0.001), but so was the number of oocytes retrieved (*p* = 0.001). However, the euploid rate and blastocyte formation rate were not significantly different, which is consistent with other studies previously cited in this review. Overall, this study shows there is no significant impact on embryo formation in those with premature progesterone elevation.

Furthermore, the morphology of embryos from cycles with premature progesterone elevation was inspected to assess for any differences or abnormalities. Hofmann et al. [79] previously described a similar morphologic embryo grade between those with and without premature progesterone elevation, by evaluating blastomere size and the presence of cytoplasmic fragments or blebs. Similarly, Hill et al. [85] reported that good-quality embryos were obtained from cycles with premature progesterone elevation; however, they did not specify the criteria for a “good” -quality embryo [85].

More recently, to better identify embryos with optimal implantation potential and improved IVF cycle outcomes, top quality embryo (TQE) characteristics were defined. TQEs entail day 2 embryos, with four equal-sized cells and no cytoplasmic fragments, which progress into day 3 embryos with eight equal-sized cells and no cytoplasmic fragments. In these studies, there are contradictory findings on the association between progesterone levels and TQEs. Baldini et al. [82] and Pardinas et al. [86] found comparable rates of TQEs in those with and without premature progesterone elevation; however, two other studies observed a significant decrease in the TQE rate in those with premature progesterone elevation [40,43].

In their retrospective study, Pardinas et al. [86] studied clinical data of 1597 patients undergoing IVF with PGT-A. Subjects underwent a GNRH antagonist gonadotropin (recombinant FSH, hMG) OS protocol, followed by an hCG trigger for final oocyte maturation. ICSI was performed on mature oocytes, and embryos were hatched on day 3 and cultured until day 5 or 6 (Table 3). Subjects were divided into two groups based on progesterone levels, <1.5 ng/mL (*n* = 1465) or ≥1.5 ng/mL (*n* = 132), on the day of hCG administration. Assessment of embryo quality was based on morphology, such as the degree of blastocyst expansion (i.e., cavitation, full expansion, or hatching out of the zona), and the quality of the inner cell mass and trophectoderm. Due to the fewer subjects with elevated progesterone levels compared to normal controls, propensity score matching was performed with 36 participants in each group. Biopsy rate, defined as the number of embryos for each biopsy per number of retrieved mature oocytes (*p* = 0.3570), TQE rate (*p* = 0.338), and number of euploid embryos (*p* = 0.958) based on PGT-A, were similar in both groups. The investigators also assessed the impact of age on embryo quality. They reported a decrease in biopsy rate by 4% (*p* < 0.01), TQE rate by 5% (*p* = 0.008), and number of euploid embryos by 10% (*p* = 0.008), for every yearly increase in subject age. These results show that serum progesterone levels did not influence oocyte and embryo quality, and that age impacts oocyte and embryo quality irrespective of serum progesterone levels in the late follicular phase.

Conversely, Huang et al. [40] performed a retrospective cohort analysis including 4236 IVF cycles with 2639 fresh ET. They incorporated the GNRH agonist protocol for pituitary desensitization, recombinant FSH for OS, and a recombinant hCG trigger for final oocyte maturation (Table 3). Subjects were divided into five different groups based on progesterone levels, as follows: ≤1.00, 1.00–1.50, 1.50–2.00, 2.00–2.50, and >2.50 ng/mL on the day of hCG administration. The main aim of this study was to analyze TQE. TQE rate, defined as number of TQE per zygote with two pronuclei, was significantly lower in subjects with serum progesterone levels > 2 ng/mL. This investigation also identified an inverse relationship between TQE rate and the patient’s duration of infertility. The findings from this study suggest that a higher progesterone concentration at the time of hCG administration results in a negative impact on embryo development.

**Table 3 cells-11-01405-t003:** Assessing the impact of progesterone levels on oocyte and embryo quality.

Study/Study Type	ART Cycle	Inclusion Criteria	Exclusion Criteria	Pituitary Desensitization Protocol	Ovarian Stimulation Protocol	Trigger	Fertilization Method	ET/Luteal Support	Assay Coefficients of Variability	Serum Progesterone Level
Hofmann et al., 1993 [79]/retrospective	IVF/ET with donor oocyte in those with ovarian failure (*n* = 68)	Subjects undergoing OS as ovum donors and subjects with ovarian failure	Not described	GNRH agonist (leuprolide acetate)	hMG	hCG	Insemination	Oral estradiol, vaginal or IM P	Intra-assay: 4.7% Inter-assay: 7.9%	Control: ≤0.9 ng/mL throughout follicular phase (*n* = 47) Experimental: >1.1 ng/mL on or day before hCG (*n* = 21)
Neves et al., 2021 [87]/retrospective	IVF cycles with frozen ET and PGT-A (*n* = 1495)	18–40-y/o infertile subjects with ICSI cycles and PGT-A	Oocyte donation cycles, conventional IVF and ICSI performed in the same cycle, use of testicular spermatozoa, fresh ET cycles, known chromosomal rearrangements or monogenic diseases	GNRH antagonist	FHS, hMG	hCG	ICSI	Not described	Intra-assay: <7%	Control: ≤1.5 ng/mL on day of trigger (*n* = 1328) Experimental: >1.5 ng/mL (*n* = 167)
Pardinas et al., 2021 [86]/retrospective	IVF cycles with PGT-A (*n* = 1597)	Subjects undergoing PGT-A	No exclusion criteria	GNRH antagonist	FSH, LH, hMG	hCG	ICSI	Not described	Intra-assay: 1.2–11.8% Inter-assay: 3.6–23.1%	Control: <1.5 ng/mL on day of hCG (*n* = 1465) Experimental: ≥1.5 ng/mL (*n* = 132)
Huang et al., 2016 [40]/retrospective	IVF (*n* = 4236) with fresh ET (*n* = 2639)	Not described	ICSI cycles, donor cycles	Long GNRH agonist (Decapeptyl and Diphereline)	rFSH (Gonal-F or Puregon)	rhCG (Ovidrel)	Insemination	Not described	Not described	Group 1: 1.00 ng/mL on day of hCG Group 2: 1.00–1.50 ng/mL Group 3: 1.50–2.00 ng/mL Group 4: 2.00–2.50 ng/mL Group 5: >2.50 ng/mL
Hernandez-Nieto et al., 2021 [83]/retrospective	IVF with PGT-A (*n* = 5141)	Only GNRH antagonist protocol	Not described	Flexible GNRH antagonist (cetrorelix acetate or ganirelix acetate)	rFSH, hMG	hCG, leuprolide acetate	ICSI	Oral estradiol, IM P	Not described	Control: ≤2 ng/mL on day of trigger (*n* = 4925) Experimental: >2 ng/mL (*n* = 216)
Fanchin et al., 1997 [35]/retrospective	IVF with fresh ET (*n* = 153)	Not described	Abnormalities of uterine cavity, abnormal sperm analysis	GNRH agonist (leuprolide acetate)	hMG	hCG	Insemination	Not described	Intra-assay: 8% Inter-assay: 11%	Control: ≤0.9 ng/mL on day of trigger (*n* = 112) Experimental: >1.1 ng/mL (*n* = 41)
Hill et al., 2015 [85]/retrospective	IVF with fresh ET (*n* = 1620)	Subjects undergoing fresh autologous ET cycles and measures serum P on day of hCG trigger	ICSI cycles, donor cycles	GNRH-antagonist (ganirelix) or GNRH-agonist (leuprolide acetate)	rFSH, hMG	hCG	Insemination or ICSI	Not described	Intra-assay: 6.7% Inter-assay: 7.2%	Group 1: ≤1.5 ng/mL on day of trigger (*n* = 1466) Group 2: >1.5 ng/mL (*n* = 114) Group 3: >2.0 ng/mL (*n* = 40)

OS, ovarian stimulation; hMG, human menopausal gonadotrophin; hCG, human chorionic gonadotropin; IM, intramuscular; P, progesterone; FSH, follicle-stimulating hormone; LH, luteinizing hormone; ICSI, intracytoplasmic sperm injection; rFSH, recombinant follicle-stimulating hormone; rhCG, recombinant human chorionic gonadotropin; PGT-A, preimplantation genetic testing for aneuploidies.

### 4.3. Clinical Outcome

The impact of premature progesterone elevation on pregnancy was also analyzed. Interestingly, results differ between those who underwent fresh vs. frozen ET. Specifically, those with elevated serum progesterone levels during oocyte retrieval cycles who underwent frozen ET, and the recipients who received donor oocytes from donor cycles with elevated serum progesterone levels, had similar, if not increased, rates of implantation, deliveries per transfer, pregnancy, and ongoing pregnancy, and similar miscarriage rates [79,82,83,84]. Additionally, there was no difference in cumulative live birth rates, i.e., the number of live-born deliveries from one ART cycle, between those with elevated vs. normal progesterone levels that underwent frozen embryo transfers and PGT-A [87]. In the studies investigating fresh ET, there were decreased, if not similar, rates of pregnancy and ongoing pregnancy, and live birth rates [44,88].

More recently, Hernandez-Nieto et al. [83] designed a retrospective cohort analysis of 5141 IVF cycles and the resulting 5806 frozen ETs. Subjects were divided into two groups based on cutoff serum progesterone levels of ≤2 ng/mL or >2 ng/mL on the day of hCG trigger. All participants underwent a GNRH antagonist protocol, which included OS with recombinant FSH and hMG, and a recombinant hCG or GNRH agonist (leuprolide acetate) trigger for final oocyte maturation if there was concern regarding ovarian hyperstimulation syndrome. Oocytes were retrieved 36 h post-trigger, and mature oocytes underwent ICSI. Embryos were graded, biopsied, and cryopreserved. Prior to thawing and embryo transfer, each subject received oral estradiol, and then IM progesterone once the endometrial lining was satisfactory (Table 3). Of the 5806 frozen ET cycles, 5617 ETs included oocytes from cycles with normal serum progesterone levels vs. 189 ETs that included oocytes retrieved from cycles with elevated serum progesterone levels. In cycles with normal vs. elevated serum progesterone levels, the rates of implantation (71.5% vs. 70.9%, *p* = 0.92), clinical pregnancy (82.3% vs. 76.2%, *p* = 0.11), ongoing pregnancy (72.1% vs. 67.8%, *p* = 0.65), multiple pregnancy (2.0% vs. 1.4%, *p* = 0.67), and clinical pregnancy loss (10.2% vs. 8.4%, *p* = 0.65) were comparable. Furthermore, there was no significant difference between gestational week at delivery (39.0 ± 1.9 vs. 39.2 ± 1.5, *p* = 0.25) and birth weight of the registered live births, in those pregnancies resulting from embryos from normal serum progesterone levels and elevated serum progesterone levels on the day of hCG administration, respectively. The investigators concluded that clinical outcomes were not impacted in those with embryos from cycles with elevated serum progesterone levels who underwent frozen ET.

Fanchin et al. [42] designed a retrospective study on 131 women who underwent 153 IVF/ETs. OS protocol included GNRH agonist (leuprolide acetate) for pituitary desensitization, hMG for OS, and an hCG trigger for final oocyte maturation. Subjects were separated into two groups based on serum progesterone level on the day of hCG trigger (≤0.9 ng/mL, *n* = 112, vs. >0.9 ng/mL, *n* = 41). Oocytes were retrieved 35 h after trigger, and inseminated 3 h later. Embryos were cultured and transferred when they reached the blastocyst stage (Table 3). In a comparison between those with elevated serum progesterone vs. normal levels at the time of trigger in fresh ET cycles, there were lower implantation rates (18 vs. 7, *p* < 0.02), clinical pregnancy rates (29 vs. 9, *p* < 0.02), and ongoing pregnancy rates (34 vs. 14, *p* < 0.03). However, there was no difference in the rate of blastocyst formation. These findings support the hypothesis that elevated serum progesterone levels in the follicular phase have minimal impact on oocyte and embryo quality. The impact on clinical outcomes of fresh ET presumably reflects the influence of elevated serum progesterone levels on the endometrium.

### 4.4. Limitations

There are limitations within the studies investigating the effects of premature progesterone elevation on oocyte and embryo quality. For example, the studies used different levels to define elevated serum progesterone on the day of ovulation trigger; different assays were used to measure progesterone; responses to OS varied among study subjects; individual study sites used specific OS and IVF/ET protocols; demographic characteristics, such as smoking history and race, were not considered; and different markers for oocyte and embryo quality were used. Additionally, compared to those with normal serum progesterone levels, most studies had far fewer participants with elevated progesterone levels, thus limiting the power of the study. Despite attempts to compensate for such discrepancies, these limitations make it difficult to generalize results to all IVF/ET cycles.

To better understand the varied results reported in studies investigating the effects of premature progesterone elevation on oocyte and embryo quality, Hill et al. [85] conducted a retrospective cohort study of 1625 fresh autologous IVF cycles, including 686 cleavage and 934 blastocyst stage ET. In this study, two values were used as the cut off for elevated serum progesterone levels (>2 ng/mL, *n* = 40 and >1.5 ng/mL, *n* = 114). The IVF protocol included GNRH antagonist or agonist (leuprolide acetate) pituitary suppression, OS using recombinant FSH and hMG, and GNRH agonist or hCG for final oocyte maturation. Oocyte retrieval was performed 36 h post-trigger, followed by ICSI or standard insemination, as clinically indicated. If there were a satisfactory number of high-quality embryos, day 3 or day 5 embryos were transferred (Table 3). A comparison of serum progesterone level and markers of oocyte and embryo quality found no associations. However, there were significant associations between patient’s age, embryo stage at time of transfer, embryo quality, number of transferred embryos, level of serum progesterone on day of trigger, and live birth rate. Irrespective of embryo stage at time of transfer, embryo quality, patient’s age, and type of ovarian responder, the cycles with elevated serum progesterone levels were linked to lower odds of live birth. Lastly, use of the GNRH antagonist protocol compared to the use of GNRH agonist protocol was twice as likely to result in serum progesterone levels >2 ng/mL (3.2% vs. 1.6%, *p* < 0.04) and >1.5 ng/mL (12% vs. 5.8%, *p* < 0.01).

### 4.5. Conclusions

Based on various markers of oocyte and embryo quality, the data strongly support the hypothesis that there is minimal to no negative influence of premature progesterone elevation on oocyte and embryo quality. Rather, the negative association between elevated progesterone levels and the clinical outcomes of fresh ET cycles is related to an endometrial effect at the molecular and histologic level. Nevertheless, there are studies showing a negative impact of elevated progesterone levels on TQE [40,43]. Thus, the question remains unanswered, and large randomized controlled trials are needed to understand the effect of elevated progesterone levels in the follicular phase on oocyte and embryo quality in IVF/ET cycles. A better understanding of this relationship will help guide recommendations on IVF protocols and ET plans for patients with premature progesterone elevation in the follicular phase, in order to ultimately improve patient outcomes.

## 5. General Conclusions

The goal of this article is to improve our understanding of the effects of elevated progesterone levels on human endometrial receptivity and oocyte/embryo quality when using assisted reproductive technologies. Despite some controversial reports, most studies, to date, strongly support the idea that a premature rise in progesterone levels in the follicular phase during ovarian stimulation is associated with reduced implantation and pregnancy rates following fresh embryo transfers (ET). This premature rise in progesterone levels largely exerts minimal or no effects on oocyte/embryo quality, while significantly advancing the histological development of the secretory endometrium and displacing the WOI. This finding strongly suggests that reduced implantation and pregnancy rates are the result of a negatively affected endometrium rather than endometrial dysfunction or poor oocyte/embryo quality. The histological evaluation of the endometrium was once viewed as the gold standard for clinical diagnosis and management of women with endometrial disorders. However, numerous studies have argued against the validity of such evaluation, mainly because of the intra- and interobserver variations in histological interpretation. Instead, it is reported that transcriptomic profiling, via endometrial receptivity arrays, offers greater and more consistent accuracy [84]. Understanding the potential negative impact of elevated progesterone levels on the endometrium is crucial for improving implantation rates following a fresh embryo transfer. Clinical studies conducted over the past three decades have greatly advanced our knowledge in this important area.

## Figures and Tables

**Figure 1 cells-11-01405-f001:**
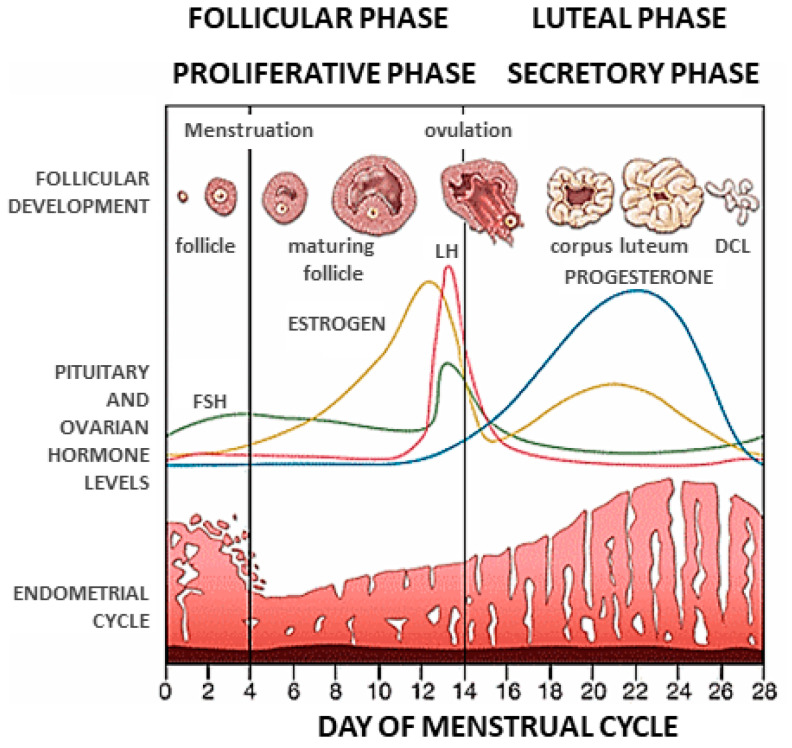
Diagram of the menstrual cycle showing the ovarian (follicular and luteal) and endometrial (proliferative and secretory) phases. The days of the menstrual cycle represent average values; durations and values may differ between different females or different cycles. DCL: degenerate corpus luteum; FSH: follicle-stimulating hormone; LH: luteinizing hormone.

**Table 1 cells-11-01405-t001:** Assessing the impact of progesterone levels on endometrial histological development.

Study/Study Type	ART Cycle	Inclusion Criteria	Exclusion Criteria	Pituitary Desensitization Protocol	Ovarian Stimulation Protocol	Trigger	ET/luteal Support	Estrogen Administration	Study Groups: Progesterone Treatment or Blood Levels	Control Groups: Progesterone Treatment or Blood Levels	Day of Estrogen and Progesterone Measurements	Day(s) of Endometrial Biopsy or Ultrasound/Method Used for Assessing Endometrial Histologic Development
Ezra et al., 1994 [48]/ prospective or retrospective was not stated	26-day artificial cycle (follicular and luteal phase) in the setting of ovarian failure	27–47-y/o women with primary or secondary premature ovarian failure, FSH and LH levels > 50 IU/L and low E levels (<25 pg/mL)	None listed	n.a.	n.a.	n.a.	n.a.	Oral micronized E: 4 mg/day over 26 days	Study Group A (*n* = 8): administered 12.5 mg P (IM in oil) on days 2 and 7 followed by 50 mg/day from day 15–26. Study Group B (*n* = 8): administered 6.25 mg P (IM in oil) on days 3, 4, and 5 followed by 50 mg/day from day 15–26	Subjects had standard preparatory cycles without follicular P supplementation (*n* = 16). Subjects only administered P (IM in oil) (50 mg/day) from luteal day 15–26	Serum E and P: days 14 and 26	14, 26/endometrial dating
Chetkowski et al., 1997 [49]/ prospective	IVF donor cycle	Healthy parous women and infertile women with functioning ovaries	None listed	GNRH agonist (leuprolide acetate) long protocol	hMG	hCG	n.a.	n.a.	n.a.	n.a.	Serum P: 1 and 2 days before hCG administration and on day of hCG administration	36 h after trigger at the time of oocyte retrieval/endometrial dating
Fanchin et al., 1999 [50]/ prospective	IVF/ET	≤38-y/o women with morphologically healthy uterus and had at least three good-quality embryos	Women whose uterine position did not allow adequate visualization of the endometrial texture at ultrasound examination and those with grossly irregular ultrasonographic appearance of the myometrium	GNRH agonist (leuprolide acetate) long protocol	hMG	hCG	ET performed 2 days after oocyte retrieval/300 mg micronized P administered daily (100 mg in the morning and 200 mg in the evening) starting on the evening of the day of ET	n.a.	Study Group (*n* = 26): plasma P > 0.9 ng/mL	Control Group (*n* = 33): plasma P ≤ 0.9 ng/mL	Plasma E and P: day of hCG administration, day of oocyte retrieval, and day of ET	Ultrasound conducted on day of hCG administration, oocyte retrieval and ET/ultrasound-based endometrial echogenicity
Liu et al., 2015 [51]/prospective	IVF without ET	23–40-y/o women with tubal or male infertility	None listed	GNRH agonist (Triptorelin) long protocol	FSH	hCG	P: 40 mg/day for 1 day, starting from the night of oocyte retrieval, and then at 60 mg/day for 2 days, and then at 80 mg/day for 3 days	n.a.	High P Group (*n* = 58): P = 1.7 ng/mL on the day of hCG administration and 9.5 ng/mL on hCG + 1	Normal P Group (*n* = 48): P <1.7 ng/mL on the day of hCG administration and <9.5 ng/mL on hCG + 1	E and P: 12–14 h before hCG administration and 12–14 h after hCG administration	7 days after hCG administration/endometrial dating
Young et al., 2017 [52]/prospective	Healthy women with induced experimental modeled cycles	19–34-y/o healthy women with regular intermenstrual interval between 25–35 days and no history of infertility or pelvic disease	An intermenstrual interval that varied by >3 days, use of medication that affects reproductive hormones or fertility within 60 days prior to enrollment, chronic disease, a body mass index >29.9 or <18.5, and history of infertility	GNRH agonist (leuprolide acetate) long protocol	n.a.	n.a.	n.a.	Transdermal 0.2 mg/day for 20 days following pituitary–ovary desensitization	P administered (IM in oil) daily after 10 days of E treatment. Group A (*n* = 6): P = 2.5 mg/day; Group B (*n* = 6): P = 5.0 mg/day; Group C (*n* = 12): 10.0 mg/day; Group D (*n* = 12): P = 40.0 mg/day	Control group (*n* = 10): natural cycles exhibiting normal reported levels of P (Nadji et al., 1975) [6]	Serum P in modeled cycles: 2–3 h after injection (peak) and 1–2 h before injection (trough) on two separate occasions between 3 and 10 days of P treatment	Control group: 10 days after the mid-cycle urinary LH surge. Modeled cycles: on P day 10 in subjects receiving 2.5 or 5 mg of P daily and from subjects receiving 10 or 40 mg of P daily/endometrial dating

E, estrogen; FSH, follicle-stimulating hormone; GNRH, gonadotropin-releasing hormone; hCG, human chorionic gonadotropin; hMG, human menopausal gonadotropin; IM, intramuscular; IVF/ET, in vitro fertilization/embryo transfer; LH, luteinizing hormone; P, progesterone; y/o, year-old; n.a., not applicable.

**Table 2 cells-11-01405-t002:** Assessing the impact of progesterone levels on the endometrial transcriptome and receptivity.

Study/Study Type	ART Cycle	Inclusion Criteria	Exclusion Criteria	Pituitary Desensitization Protocol	Ovarian Stimulation Protocol	Trigger	ET/Luteal Support	Estrogen Administration	Study Groups: Progesterone Treatment or Blood Levels	Control Groups: Progesterone Treatment or Blood Levels	Day of Estrogen and Progesterone Measurements	Day(s) of Endometrial Biopsy or Ultrasound/Method Used for Assessing Endometrial Transcriptome
Young et al., 2017 [52]/prospective	Healthy women with induced experimental modeled cycles	19–34-y/o healthy women with regular intermenstrual interval between 25–35 days and no history of infertility or pelvic disease	Intermenstrual interval that varied by >3 days, use of medication that affect reproductive hormones or fertility within 60 days prior to enrollment, chronic disease, a body mass index >29.9 or <18.5, and history of infertility	GNRH agonist (leuprolide acetate) long protocol	n.a.	n.a.	n.a.	Transdermal 0.2 mg/day for 20 days following pituitary–ovary desensitization	P administered (IM in oil) daily after 10 days of E treatment. Group A (*n* = 6): P = 2.5 mg/day; Group B (*n* = 6): P = 5.0 mg/day; Group C (*n* = 12): 10.0 mg/day; Group D (*n* = 12): P = 40.0 mg/day	Control group (*n* = 10): natural cycles exhibiting normal reported levels of P (Nadji et al., 1975) [6]	Serum P in modeled cycles: 2–3 h after injection (peak) and 1–2 h before injection (trough) on two separate occasions between 3 and 10 days of P treatment	Control group: 10 days after the mid-cycle urinary LH surge (cycle day 23 in the mid-luteal phase). Modeled cycles: on P day 10 in subjects receiving 2.5 or 5 mg of P daily and from subjects receiving 10 or 40 mg of P daily/microarray hybridization with RT- PCR and in silico comparison to previous studies
Labarta et al., 2011 [66]/ prospective	OS without IVF (oocyte donors)	18–350-y/o women, body mass index 18–25, 25–35-day menstrual cycles, normal basal serum hormone levels on day 3 of the menstrual cycle (FSH < 10 IU/L, LH < 10 IU/L, and E2 < 60 pg/mL, normal karyotype	Endometriosis, polycystic ovarian syndrome	GNRH agonist (leuprolide acetate) long protocol or GnRH antagonist (Cetrotide)	rFSH	hCG	No ET performed luteal support with 400 mg/day of micronized P administered 1 day after oocyte retrieval to simulate ET cycles	n.a.	High P Group (*n* = 6): P > 1.5 ng/mL on the day of hCG trigger	High P Group (*n* = 6): P > 1.5 ng/mL on the day of hCG trigger	Serum E2 on day 3 of menstrual cycle Serum P on day of hCG administration	7 days after hCG administration (hCG + 7)/microarray hybridization
Haouzi, et al., 2014 [67]/ prospective	IVF/ET	28–34-y/o women, male factor infertility, normal serum FSH, LH, and E2 on day 3 of OS and on the day of hCG administration	No specific criteria listed	GNRH agonist long protocol or GnRH antagonist	hMG	hCG	ET performed 3 days after oocyte retrieval/luteal support not specified	n.a.	High P Group (*n* = 8): P > 1.5 ng/mL on the day of hCG trigger	Control Group (*n* = 7): P < 1.5 ng/mL on the day of hCG trigger	Serum E2 on day 3 of OS Serum P on day of hCG administration	Pre-receptive (hCG + 2) and receptive (hCG + 5) secretory stages/microarray hybridization with RT- qPCR
Xiong, et al., 2020 [61]/ prospective	IVF without ET	24–40-y/o women, body mass index 18–25, 25–35-day menstrual cycles, normal basal serum hormone levels on days 2–4 of the menstrual cycle (FSH < 10 IU/L and E2 < 60 pg/mL, tubal or male factor infertility, normal karyotype in both partners	Polycystic ovarian syndrome, hydrosalpinx, uterine abnormalities, thyroid dysfunction, recurrent miscarriage	GNRH agonist super-long or long protocol	FSH	hCG	No ET performed/luteal support not received	n.a.	High P Group (*n* = 20): P ≥ 1.7 ng/mL on day of hCG administration	Normal P Group (*n* = 20): P < 1.7 ng/mL on the day of hCG administration	Serum E2 on days 2–4 of menstrual cycle Serum P every 2–4 days during OS and on day of hCG administration	7 days after hCG administration/Sequenom MALDI-TOF mass spectrometry or bisulfate sequencing PCR, and immunohistochemistry

E2, estradiol; IM, intramuscular; P, progesterone; n.a., not applicable; hCG, human chorionic gonadotropin; hMG, human menopausal gonadotropin; GnRH, gonadotropin-releasing hormone; FSH, follicle-stimulating hormone; IVF/ET, in vitro fertilization/embryo transfer; rFSH, recombinant follicle-stimulating hormone; y/o, year-old.

## Data Availability

Not applicable.

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
