# Peer review of "Towards an Improved Understanding of the Effects of Elevated Progesterone Levels on Human Endometrial Receptivity and Oocyte/Embryo Quality during Assisted Reproductive Technologies"

_cells, 2022, doi:10.3390/cells11091405_

Round 1

Reviewer 1 Report

The current review by Kalakota and col. includes a comprehensive summary of the state of the art on progesterone levels and the related outcomes on oocyte and embryo quality during ART. Moreover, authors conclude, based on their exhaustive research, that endometrium histological changes and/or receptivity migth me definitely link to the reduce ART outcomen whemn it comes to elevate progesterone in woment with fertility issues.

Minor issues:

Pag 13 of 26. LIN74-87. Please, include some of the genes that belong to cell adhesion, developmental and immune processes, for better clarification for the readers.

Author Response

Response to Reviewer 1 Comments

Point 1: Minor issues: Page 13 of 26. LIN74-87. Please, include some of the genes that belong to cell adhesion, developmental and immune processes, for better clarification for the readers.

Response 1: To better clarify for the reader, I changed my phrasing to include two examples of genes involved in each of the biological processes mentioned. The phrasing now reads as following, “analyses of the biological processes, molecular functions and KEGG pathways associated with the differentially expressed genes revealed differential expression among genes involved in cell adhesion (examples, LAMA4 and ITGB2), as well as developmental (examples, SMAD9 and RND3) and immune processes (examples, ILIB and TLR5).

Reviewer 2 Report

This manuscript contains a narrative report of "selected studies spanning three decades of clinical research". There is no explanation on the selection criteria of the reports included. The narrative report is divided into chapters devoted to various aspects of the research published and presents in a clear and intelligent way the rational behind its conclusions, including a meticulous description of the limitations of the studies, helping the reader to make a valuable consideration of the findings. This review presents a clear description of the studies indicating the correlation of lower implantation and pregnancy rates and premature elevation of progesterone following COH for IVF-ET. It presents convincingly the case for embryo-endometrial asynchrony. This review does not contain the studies reporting on the critical level of progesterone that is detrimental (for example see publications by Fatemi HM) nor recent publication showing no effect of elevated P on embryo ploidy rate (Neves et al, Reprod Biomed Online. 2021 Dec;43(6):1063-1069). Still, basically it is a nicely written narrative overview of some of the pioneering reserch regarding the topic, leading to sound conclusions and therefore merits publication. 

Author Response

Response to Reviewer 2 Comments

Point 1: This manuscript contains a narrative report of "selected studies spanning three decades of clinical research". There is no explanation on the selection criteria of the reports included.

Response 1: To clarify the selection criteria, the following explanation is included in our general introduction on page 3 lines 125-129: This article reviews published clinical studies from PubMed spanning the period 1994-2021. Studies were first identified using keyword search terms that included: ovarian, stimulation, progesterone, gene expression, oocyte, embryo and development. Only studies reporting a clearly defined elevated progesterone level, as the single variable factor following ovarian stimulation, were included in this review. Additionally, we have improved Tables 1-3 and now include the inclusion and exclusion criteria for each study, which the reader is asked to refer to in the main text. 

Point 2: This review does not contain the studies reporting on the critical level of progesterone that is detrimental (for example see publications by Fatemi HM) nor recent publication showing no effect of elevated P on embryo ploidy rate (Neves et al, Reprod Biomed Online. 2021 Dec;43(6):1063-1069). Still, basically it is a nicely written narrative overview of some of the pioneering research regarding the topic, leading to sound conclusions and therefore merits publication. 

Response 2: To address the effect that elevated progesterone may have on embryo ploidy rate, the suggested study by Neves et al., 2021 was reviewed on pages 21-22 lines 307-318. The following explanation was provided in the main body of the text “when looking at embryo ploidy rates (Neves et al., 2021) conducted a multicenter retrospective study of 1495 IVF cycles with ICSI that underwent preimplantation genetic testing for aneuploidy (PGT-A). A GNRH antagonist protocol (recombinant FSH, hMG) for OS was used and followed by an hCG trigger for oocyte maturation. Embryos were cultured and underwent a freeze all strategy after testing (Table 3). Subjects were divided into two groups based on serum progesterone level on the day of hCG trigger: ≤ 1.50 ng/mL (n=1328) and > 1.50 ng/mL (n=167). Interestingly, in the progesterone elevation group, the number of euploid embryos was significantly higher (p = 0.001), but so was the number of oocytes retrieved (p = 0.001). However, the euploid rate and blastocyte formation rate were not significantly different which is consistent with other studies previously cited in this review. Overall, this study shows there is no significant impact on embryo formation in those with premature progesterone elevation.” Additionally, based on findings from several studies cited on page 21 line 306, no differences were observed in duration of embryo culture, cleavage rate, and blastocyst rate when cycles with and without premature progesterone elevation were compared

Reviewer 3 Report

I read the article entitled ‘Towards an improved understanding of the effects of elevated progesterone levels on human endometrial receptivity and oo-cyte/embryo quality during assisted reproductive technologies’

The manuscript was very well written, but it was very complex and difficult to read.

The article must be reconstructed according to ‘ PRISMA 2020 statement comprises a 27-item checklist addressing the introduction, methods, results and discussion sections of a systematic review report.’

It should be applied for all three parts of manuscript.

  1. The Impact of Progesterone Levels on Endometrial Histological Development And Endometrial Receptivity
  2. The Impact of Progesterone Levels on The Endometrial Transcriptome And Endometrial Receptivity
  3. The Impact of Elevated Progesterone Levels on Oocyte And Embryo Quality

Ref: 1. BMJ (OPEN ACCESS) Page MJ, McKenzie JE, Bossuyt PM, Boutron I, Hoffmann TC, Mulrow CD, et al. The PRISMA 2020 statement: an updated guideline for reporting systematic reviews. BMJ 2021;372:n71. doi: 10.1136/bmj.n71

Author Response

Response to Reviewer 3 Comments

Point 1: The manuscript was very well written, but it was very complex and difficult to read.

Response 1: Overall the reviewers agree that this is a well writing manuscript. The detail in which the referenced studies are described may contribute to the complexity of the review. However, the manuscript was reviewed and edited to improve clarity and organization.

Point 2: The article must be reconstructed according to ‘ PRISMA 2020 statement comprises a 27-item checklist addressing the introduction, methods, results and discussion sections of a systematic review report.’

It should be applied for all three parts of manuscript.

  1. The Impact of Progesterone Levels on Endometrial Histological Development And Endometrial Receptivity
  2. The Impact of Progesterone Levels on The Endometrial Transcriptome And Endometrial Receptivity
  3. The Impact of Elevated Progesterone Levels on Oocyte And Embryo Quality

Response 2: After carefully reviewing the PRISMA 2020 guidelines, we find that most of the PRISMA guidelines are not applicable to our review. PRISMA deals with the formatting of systematic reviews, however our manuscript is a literature review. It is not possible for us to reformat our manuscript according to PRISMA 2020 as there are also important differences outlined here https://venebio.com/news/2017/09/5-differences-between-a-systematic-review-and-other-types-of-literature-review/.  Given this concern, the editorial office was contacted for clarification, and we received the following response from the academic editors “I would suggest that the authors address reviewer 3 comments based on the logistic of the literature review, not a systemic review like what they’ve mentioned in the email you attached. However, I suggest that authors include criteria (also asked by another reviewer) whenever application. In addition, I agree that the review is very dense, the authors should consider reorganizing the writing if possible for a more concise review.” After receiving this feedback our manuscript was revised to include several things listed in the PRISMA guidelines, such as stating the inclusion and exclusion criteria for each study in tables 1, 2 and 3.  Additionally, the manuscript was revised to improve organization and brevity.

Reviewer 4 Report

During IVF, several ovarian follicles are produced by ovarian stimulation. Abnormally high levels of estradiol and a premature increase in progesterone can be observed on the day of hCG administration. These hormone levels are linked to decreased embryo implantation and pregnancy rates in fresh embryo transfer cases.

This review focuses on the effects of high progesterone levels on the endometrium and the quality of transplanted oocytes and embryos, from three years of selected clinical studies.

This very well conducted review is useful and deserves to be published. the manuscript nevertheless requires some improvements. Here are some remarks.

Introduction

A figure or table summarizing the menstrual cycle with variations in hormone levels and effects on the endometrium would be helpful.

Page 2, line 94: How were the articles chosen? From Pubmed? other databases? In a classic way such as collecting references published for 30 years? Were there any keywords? Which?

A: The impact of progesterone levels on endometrial histological development and endometrial receptivity

The studies referenced in this section help to understand the effects of high levels of progesterone on the histological characteristics of the endometrium and, consequently, on its receptivity.

Page 4, table 1: It would be useful to give all the abbreviations in the legend, even if they are in the text, which would facilitate the reading of the tables. Also check if the abbreviations are all given in the text, check if the abbreviations given in the legend are indeed in the table.

Page 6, line 13: replace “histologic” with “histological”.

 B: The Impact of progesterone levels on the endometrial transcriptome and endometrial receptivity

In this section, a review of several studies shows that there is a link between progesterone levels and endometrial changes at the transcriptomic level, but there are also limitations that need to be considered.

Page 10, line 239: replace “histologic” with “histological”.

Page 11, table 2: Like in table 1, it would be useful to give all the abbreviations in the legend, even if they are in the text, which would facilitate the reading of the tables. Also check if the abbreviations are all given in the text, check if the abbreviations given in the legend are indeed in the table.

C: The impact of elevated progesterone levels on oocyte and embryo quality

According to current pathophysiological knowledge of the menstrual cycle and early pregnancy, it appears that a premature rise in progesterone levels would impact the receptivity of the endometrium and the quality of oocytes and embryos after ovarian stimulation. However, the results of the studies listed in this section present contradictions.

Page 18, table 3: like in the other tables, it would be useful to give all the abbreviations in the legend, even if they are in the text, which would facilitate the reading of the tables. Also check if the abbreviations are all given in the text, check if the abbreviations given in the legend are indeed in the table.

 General conclusion

A point that seems important in the conclusion is that the histological evaluation of the endometrium, once considered the gold standard for the diagnosis and management of patients, has now been challenged by numerous studies. These variations are mainly related to variations in the histological interpretation depending on the observer. Transcriptomic profiling would provide greater accuracy and more consistent results.

Abbreviations

Page 22. This directory of abbreviations is very useful to the reader because there are many of them and it is not always easy to refer to the content of the text or the tables. Check carefully if all the abbreviations are indicated.

References

Check carefully the references.

Reference [90] is given in table 3, but I didn’t find itb in the text: is it normal?

Author Response

Response to Reviewer 4 Comments

Point 1: Introduction: A figure or table summarizing the menstrual cycle with variations in hormone levels and effects on the endometrium would be helpful.

Response 1: To demonstrate the fluctuations in hormone levels during a menstrual cycle, Figure 1 was included on page 2 and is referred to at several places within the body of the Introduction.

The figure is a composite of two Creative Commons figures obtained though Microsoft's Power Point.

Point 2: Page 2, line 94: How were the articles chosen? From Pubmed? other databases? In a classic way such as collecting references published for 30 years? Were there any keywords? Which?

Response 2: To clarify the selection criteria, this explanation was included in our general intro page 3 lines 125-129: This article reviews published clinical studies from PubMed spanning the period 1994-2021. Studies were first identified using keyword search terms that included: ovarian, stimulation, progesterone, gene expression, oocyte, embryo and development. Only studies reporting a clearly defined elevated progesterone level, as the single variable factor following ovarian stimulation, were included in this review.

Point 3: Page 4, table 1: It would be useful to give all the abbreviations in the legend, even if they are in the text, which would facilitate the reading of the tables. Also check if the abbreviations are all given in the text, check if the abbreviations given in the legend are indeed in the table.

Response 3: The table legend was revised to include the abbreviations. The main text was reviewed, and the abbreviation list was updated to include all abbreviations used. 

Point 4: Page 6, line 13: replace “histologic” with “histological”.

Response 4: This replacement was made.

 Point 5: Page 10, line 239: replace “histologic” with “histological”.

Response 5: This replacement was made.

Point 6: Page 11, table 2: Like in table 1, it would be useful to give all the abbreviations in the legend, even if they are in the text, which would facilitate the reading of the tables. Also check if the abbreviations are all given in the text, check if the abbreviations given in the legend are indeed in the table.

Response 6: The table legend was revised to include the abbreviations. The main text was reviewed, and the abbreviation list was updated to include all abbreviations used. 

Point 7: Page 18, table 3: like in the other tables, it would be useful to give all the abbreviations in the legend, even if they are in the text, which would facilitate the reading of the tables. Also check if the abbreviations are all given in the text, check if the abbreviations given in the legend are indeed in the table.

Response 7: The table legend was revised to include the abbreviations. The main text was reviewed, and the abbreviation list was updated to include all abbreviations used. 

Point 8: Page 22. This directory of abbreviations is very useful to the reader because there are many of them and it is not always easy to refer to the content of the text or the tables. Check carefully if all the abbreviations are indicated.

Response 8: The main text was reviewed, and the abbreviation list was updated to include all abbreviations used. 

Point 9: Reference [90] is given in table 3, but I didn’t find itb in the text: is it normal?

Response 9: The reference list was reviewed to confirm reference 90 was included.  

Round 2

Reviewer 1 Report

The current version of the manuscript fulfills, in my opinion, merits to be published in Cells-journal.

Author Response

Thank you for the feedback. 

Reviewer 3 Report

Dear Authors,

The article is suitable for publication after revision.

Yours sincerely,

Author Response

Thank you for your feedback. 

Reviewer 4 Report

Comments on the revised version (V2) of the manuscript:

“Towards an improved understanding of the effects of elevated progesterone levels on human endometrial receptivity and oocyte/embryo quality during assisted reproductive technologies”

Thanks to the authors for taking my comments into account. For me, this manuscript can be published after a final improvement: carefully check the references in order to present them according to the standard of the journal. In the text, references are now spelled out instead of numbered, and I found both spelled references and numbered references (example: page 1, lines 37 and 43 or page 11 , lines 219 and 222).
